# Fluidity of Poly (ε-Caprolactone)-Based Material Induces Epithelial-to-Mesenchymal Transition

**DOI:** 10.3390/ijms21051757

**Published:** 2020-03-04

**Authors:** Sharmy Saimon Mano, Koichiro Uto, Mitsuhiro Ebara

**Affiliations:** 1International Center for Materials Nanoarchitectonics (WPI-MANA), National Institute for Materials Science (NIMS), 1-1 Namiki, Tsukuba 305-0044, Ibaraki, Japan; sharmy.simon@gmail.com; 2International Center for Young Scientist (ICYS), National Institute for Materials Science (NIMS), 1-1 Namiki, Tsukuba 305-0044, Ibaraki, Japan; uto.koichiro@nims.go.jp; 3Graduate School of Pure and Applied Sciences, University of Tsukuba, 1-1-1 Tennodai, Tsukuba 305-8577, Ibaraki, Japan; 4Graduate School of Tokyo University of Science, 6-3-1 Niijuku, Katsushika-City 125-8585, Tokyo, Japan

**Keywords:** fluidity, poly (ε-caprolactone), dynamic material, epithelial to mesenchymal transition, mechanotransduction

## Abstract

Background: We propose the potential studies on material fluidity to induce epithelial to mesenchymal transition (EMT) in MCF-7 cells. In this study, we examined for the first time the effect of material fluidity on EMT using poly(ε-caprolactone-*co*-D,L-lactide) (P(CL-*co*-DLLA)) with tunable elasticity and fluidity. Methods: The fluidity was altered by chemically crosslinking the polymer networks. The crosslinked P(CL-*co*-DLLA) substrate showed a solid-like property with a stiffness of 261 kPa, while the non-crosslinked P(CL-*co*-DLLA) substrate of 100 units (high fluidity) and 500 units (low fluidity) existed in a quasi-liquid state with loss modulus of 33 kPa and 30.8 kPa, respectively, and storage modulus of 10.8 kPa and 20.1 kPa, respectively. Results: We observed that MCF-7 cells on low fluidic substrates decreased the expression of E-cadherin, an epithelial marker, and increased expression of vimentin, a mesenchymal marker. This showed that the cells lose their epithelial phenotype and gain a mesenchymal property. On the other hand, MCF-7 cells on high fluidic substrates maintained their epithelial phenotype, suggesting that the cells did not undergo EMT. Conclusion: Considering these results as the fundamental information for material fluidity induced EMT, our system could be used to regulate the degree of EMT by turning the fluidity of the material.

## 1. Introduction

The epithelial cell layer is a membranous tissue that forms the covering of the internal and external surfaces of organs to protect them from their environment. It also acts as a fundamental cell layer to segregate different compartments of the organism, transport materials through the compartments, and produce extracellular components [1]. The interaction of epithelial cells with the extracellular matrix (ECM) directs diverse cellular processes such as proliferation [2], migration [3], and differentiation [4]. These cells bind with ECM through integrin-based proteins known as focal adhesion, which acts as a connector between the intracellular cytoskeleton and ECM [5]. The interaction between integrin and ECM is also critical in developmental and regeneration processes as well as in the formation of diseases.

Epithelial to mesenchymal transition (EMT) is one of such events regulated partly by ECM, in which polarized epithelial cells undergo several biochemical changes and develop a mesenchymal phenotype [6]. EMT is a major physiological process that occurs during embryogenesis, wound healing, fibrosis, and cancer metastasis [7]. The EMT process occurs through a series of events such as transcription factor activation [8], specific cell surface protein expression [9], reorganization and expression of cytoskeletal proteins [10], regulation of ECM proteins [11], degradation of cell to cell adhesion [12], regulation in the expression of specific miRNAs [13], and an increased ability to migrate and invade [6]. Further, both mechanical and biochemical cues such as ECM stiffness, tumor necrosis factor α (TGF α), nuclear factor kappa B (NF-κB), interleukin-8 (IL-8), metalloproteinase-3, and ECM proteins can regulate the degree and duration of an EMT event [14]. Moreover, serum proteins and complement component C3a stimulate the transformation of epithelial cells to mesenchymal phenotypes when cultured in a three-dimensional microfluidic chip [15]. On the other hand, it is unknown how material fluidity is directly involved in the regulation of EMT.

In our previous report, we showed that the material fluidity induces senescent type cell death in cancer cells which halt cancer metastasis [16]. In this study, we propose that the material fluidity or viscosity induce EMT in MCF-7 cells. The deficiency in E-cadherin expression and up-regulated vimentin are the key events in EMT phenomena. E-cadherin is a transmembrane glycoprotein involved in cell to cell adhesion which is essential to maintain the structural integrity of the cell [17]. Down-regulation of E-cadherin promotes cell junction loosening and increases the migration behavior of the cells [18]. Vimentin is an intermediate filament protein and its up-regulation is related to the migration of cells. Vimentin regulates cellular polarity, the cytoskeleton, cell adhesion molecules, cell contact formation, and transport of signal proteins responsible for cell motility along with the regulatory proteins [19]. However, the regulatory protein involved in EMT depends on the type of EMT phenomena and which cells underwent EMT [20].

Here, we prepared a fluidic cell culture platform using poly(ε-caprolactone-*co*-D,L-lactide) (P(CL-*co*-DLLA)). We observed that the epithelial cells underwent EMT in non-crosslinked P(CL-*co*-DLLA) 500 substrate by a gradual decrease in the expression of E-cadherin, an epithelial marker, and an increase in the expression of vimentin. The expression of transforming growth factor-β (TGF-β) is also increased with an increase in the duration of cell culture on non-crosslinked P(CL-*co*-DLLA) 500 substrates. Our results suggest that material fluidity is a crucial factor to determine the EMT process. Moreover, the induction of EMT depends on the duration of cell culture on P(CL-*co*-DLLA) substrates as well as the fluidity or viscosity. Our system could be considered as an ideal substrate to induce EMT phenomena and to study the mechanism of regulation of EMT. Further, it is possible to regulate the degree of EMT by altering the fluidity of the material.

## 2. Results

### 2.1. Characterization of Crosslinked P(CL-co-DLLA) and Non-Crosslinked P(CL-co-DLLA) Substrates.

Previously we had reported novel techniques to control the melting temperature (*Tm*) of poly(ε-caprolactone) (PCL) by tailoring the number of branched chains [21,22,23,24] for biological applications. As an alternative method, we also reported that *Tm* was adjusted below human body temperature by copolymerization of different ratios of CL with DLLA [21]. Here, we prepared four-branched copolymers with a CL/DLLA ratio of 60/40 of 100 or 500 units.

The molecular weight (Mw) and molecular number (Mn) of copolymers of 100 units were 59,600 and 40,900 (Mw/Mn = 1.46) and copolymers of 500 units were 119,142 and 50,261, respectively (Mw/Mn = 2.37) as determined by gel permeation chromatography (GPC). The CL and DLLA contents of the copolymers were determined by ^1^H-NMR to be 61.1 and 38.9 mol%, respectively, for 100 and 59.9 and 40.1 mol% for 500 when the feed concentration was 60 and 40 mol% (Appendix A, Appendix A). The viscoelastic properties of copolymers of 100 unit substrate were: Storage modulus, G’ = 10.8 kPa, loss modulus, G’’ = 33.0 kPa and tan δ (=G’’/G’) = 3.06 (>>1 means that the substrate is fluidic) and 500 unit substrate storage modulus, G’ = 20.1 kPa, loss modulus, G’’ = 30.8 kPa and tan δ (=G’’/G’) = 1.53 (Appendix A). The shear modulus (G*) of copolymers of 100 and 500 unit non-crosslinked P(CL-*co*-DLLA) substrate were 34.7 and 36.8 kPa, respectively, and were calculated from the formula √ (G’)2+ (G”)2. The tensile test showed the crosslinked P(CL-*co*-DLLA) substrate had a stiffness of 261 kPa. However, the water contact angle (Appendix Aa,b) and surface roughness (Appendix Ac) of crosslinked and non-crosslinked P(CL-*co*-DLLA) substrates remained unchanged.

### 2.2. Thickness Based Effect of Non-crosslinked P(CL-co-DLLA) Substrates on Cell Spreading

To optimize the thickness of spin-coated non-crosslinked P(CL-*co*-DLLA) substrates on cell culture, the cover glasses were spin-coated with different concentrations (0.1 *w/v%* to 100 *w/v%*) of non-crosslinked P(CL-*co*-DLLA) 100 or 500 substrates to obtain a different thickness of materials on the cover glass. To determine the thickness of the substrates on the cover glass, the substrates were mixed with the dye nile red. Appendix A shows the 3D images of the thickness of non-crosslinked P(CL-*co*-DLLA) substrates at different concentrations. The confocal microscopic image of MCF-7 cells on crosslinked P(CL-*co*-DLLA) substrate showed a well-spread cell morphology (Figure 1a). Interestingly, the cell spreading behavior of MCF-7 cells decreased with an increased concentration of non-crosslinked P(CL-*co*-DLLA) 100 substrates. MCF-7 cells showed a well-spread morphology in 1 *w/v%* and completely rounded morphology in 50 *w/v%* or higher on non-crosslinked P(CL-*co*-DLLA) 100 substrates (Figure 1a, top images). However, MCF-7 remained in their elongated morphology in any of the concentrations (1–100 *w/v%*) of non-crosslinked P(CL-*co*-DLLA) 500 substrates (Figure 1a, bottom images). This suggests that the material fluidity and thickness used for cell culture influences the cell morphology and spreading behavior. Figure 1b shows a graphical representation of the spreading behavior of the cells.

### 2.3. Effect of Non-Crosslinked P (CL-co-DLLA) Substrates on E-Cadherin and Vimentin Expression

The effects of materials stiffness on the transformation of epithelial cells to mesenchymal phenotype have been studied well [25]. On the other hand, it is unknown what the role of material fluidity is on EMT. To understand this, we cultured the cells on the crosslinked (solid-like) and non-crosslinked (liquid-like) P(CL-*co*-DLLA) 100 or 500 substrates for three days and monitored the presence or absence of epithelial and mesenchymal proteins such as E-cadherin and vimentin, respectively.

The EMT event started within 24 h in both crosslinked P(CL-*co*-DLLA) and non-crosslinked P(CL-*co*-DLLA) 500 substrates. About 10% and 40% of MCF-7 epithelial cells were transformed to a mesenchymal phenotype in crosslinked and non-crosslinked P(CL-*co*-DLLA) 500 substrates, respectively, at 24 h which was confirmed by the expression of the epithelial and mesenchymal markers such as E-cadherin and vimentin, respectively, in those cells (Appendix A). In the crosslinked P(CL-*co*-DLLA) substrate, about 65% and 75% of MCF-7 epithelial cells were transformed to mesenchymal cells at 48 h (Figure 2a,b) and 72 h (Figure 2c,d), respectively. In non-crosslinked P(CL-*co*-DLLA) 500 substrate, 48 h, about 85% of MCF-7 cells and at 72 h (Figure 2a,b) around 96% (Figure 2c,d) of the cells were transformed to the mesenchymal phenotype. This shows that the EMT event depends on time and the materials in which the cells have cultured. Our data clearly show a gradual transformation of epithelial to mesenchymal phenotype of MCF-7 cells on non-crosslinked P(CL-*co*-DLLA) 500 substrates. On the other hand, MCF-7 cells on non-crosslinked P(CL-*co*-DLLA) 100 substrates maintained their epithelial phenotype throughout the culture period (Appendix A, Figure 2).

To confirm the EMT process in crosslinked P(CL-*co*-DLLA) and non-crosslinked P(CL-*co*-DLLA) 100 and 500 substrates, we analyzed the expression of E-cadherin and vimentin mRNA across five days. On day two, MCF-7 cells on non-crosslinked P(CL-*co*-DLLA) 500 substrates showed a significant reduction in the expression of E-cadherin mRNA (Figure 2e) and an increase in the vimentin mRNA (Figure 2f) compared to the crosslinked P(CL-*co*-DLLA) substrate. After four days, the expression of E-cadherin and vimentin mRNA reached the same level in both crosslinked P(CL-*co*-DLLA) and non-crosslinked P(CL-*co*-DLLA) 500 substrates. After four days MCF-7 cells in both crosslinked P(CL-*co*-DLLA) and non-crosslinked P(CL-*co*-DLLA) 500 substrates converted to mesenchymal cells.

### 2.4. Effect of Substrate Fluidity on TGF-β Expression

TGF-β plays a principal role in the induction of EMT. TGF-β is a multifunctional cytokine that directs cell development, differentiation and homeostasis, and cellular transformation like EMT [26]. TGF-β promotes tumor cell motility and invasion through the induction of an EMT [27]. Next, we checked the expression of TGF-β mRNA to correlate with the EMT process. Interestingly, we found that there was an increase in the expression of TGF-β mRNA both in crosslinked P(CL-*co*-DLLA) and non-crosslinked P(CL-*co*-DLLA) 500 substrates (Figure 3). Initially, there was a significant difference in the expression and later the expression level reached the same. This shows that the up-regulation of TGF-β mRNA is also associated with EMT.

### 2.5. Effect of Non-Crosslinked P(CL-co-DLLA) Substrate on Cell Adhesion

First, we checked the adhesion behavior and expression of focal adhesion protein of MCF-7 cells on crosslinked and non-crosslinked P(CL-*co*-DLLA) substrates. For that, we cultured and incubated the cells for at least four hours on the substrates coated with or without fibronectin. After 4 h we checked what percentage of cells adhered to the substrates. About 83% of cells adhered to crosslinked P(CL-*co*-DLLA) and non-crosslinked P(CL-*co*-DLLA) 500 substrates. However, only 63% of the cells adhered to non-crosslinked P(CL-*co*-DLLA) 100 substrates (Figure 4a, black bars). Interestingly, there was a significant difference in cell adhesion behavior on fibronectin-coated substrates. The number of cells adhered increased to about 93%, 71%, and 86% on crosslinked P(CL-*co*-DLLA) and non-crosslinked P(CL-*co*-DLLA) 100 and 500 substrates, respectively (Figure 4a, white bars). This shows that our substrates favor the adhesion of the cells for cell culture experiments.

The expression of focal adhesion is one of the crucial factors associated with the ability of cells to move during the EMT process [28]. Thus, we checked the expression of focal adhesion protein vinculin on crosslinked and non-crosslinked P(CL-*co*-DLLA) substrates (Figure 4b, Appendix A). A visible expression of focal adhesion protein was observed on both crosslinked (Figure 4b, top images, indicated with white arrows) and non-crosslinked P(CL-*co*-DLLA) 500 substrates (Figure 4b, bottom images, indicated with white arrows). Yet, cells on non-crosslinked P(CL-*co*-DLLA) 100 substrate did not show the expression of visible focal adhesion protein (Figure 4b, middle images). This shows that the expression of focal adhesion protein is correlated with the adhesion behavior of the cells on crosslinked and non-crosslinked P(CL-*co*-DLLA) substrates.

### 2.6. Effect of Non-crosslinked P(CL-co-DLLA) Substrate on Cell Proliferation

To understand the proliferation behavior of MCF-7 cells on crosslinked or non-crosslinked P(CL-*co*-DLLA) substrates, we cultured MCF-7 cells on those substrates for five days. The cell proliferation increased by increasing the culture period on non-crosslinked P(CL-*co*-DLLA) 500 substrates. In contrast, on the non-crosslinked P(CL-*co*-DLLA) 100 substrate, the proliferation initially increased until two days. From day 3 onward, proliferation decreased. This shows that non-crosslinked P(CL-*co*-DLLA) 500 substrates favored the proliferation behavior of the cells, however, non-crosslinked P(CL-*co*-DLLA) 100 declined (Figure 5).

## 3. Discussion

Biomaterials mimicking ECM have been applied to direct numerous cellular functions for tissue engineering [29] as well as regenerative medicine [30]. The adhesion of cells to the ECM is crucial to governing cellular behavior and function. Cells adhere to the ECM via integrin-based proteins. A large group of proteins aid this phenomenon. They are crosslinked to each other and to the cell to favor cell attachment to the ECM. The initial adhesion behavior of the cell to the ECM is known as nascent adhesion and later nascent adhesion is matured and formed to focal adhesion. During cell migration, focal adhesion grasps and develops a force necessary to pull the cell body forward [31]. Furthermore, cell adhesion has been enhanced by coating the cell culture surface with the functional domain of ECM proteins such as collagen I, collagen IV, and fibronectin [32].

Surface stiffness is an imperative factor involved in the cell-material interaction that alters cell behaviors such as adhesion, morphology [33], proliferation [34] and differentiation [35,36]. Thus cell-material interaction is crucial in understanding the cellular response on these materials to the design and modification of biomaterials for biomedical applications.

Cell-ECM adhesions are key mediators of cellular mechanotransduction [37]. EMT is a biochemical process in which epithelial cells lose cell–cell adhesion and gain the properties of mesenchymal cells which show migratory and motile properties. EMT is well known for its crucial role in embryogenesis and development. Focal adhesion plays a vital role in the regulation of the EMT process. It is hypothesized that the inhibition of focal adhesion kinase (FAK) might disable the cell’s ability to sense ECM stiffness and in turn prevent stiffness or sensitive EMT. Moreover, FAK is involved in the regulation of cancer proliferation [38] and transcriptional regulation of mesenchymal markers [27]. MCF-7 cells on crosslinked and non-crosslinked P(CL-*co*-DLLA) 500 substrates show visible expression of focal adhesion proteins and promote the proliferation of cells.

Material stiffness plays a vital role in the regulation of EMT or mesenchymal to epithelial transition (MET) and vice-versa. The stiffness-based effect on polyelectrolyte-based materials promotes endothelial to mesenchymal transition (EndMT) for vascular implant [39]. MDA-MB-231 and MCF-7 cells were adapted to an elongated or spindle-shaped morphology when these cells were cultured on tantalum oxide nanochips of 100–200 nm in diameter. E-cadherin gene expression decreased and N-cadherin and vimentin expression was enhanced. This is the case when epithelial cells are cultured on a microfluidic device with a flow of healthy serum and heat-inactivated serum with C3a stimulate mesenchymal and epithelial phenotypes, respectively [40]. On the other hand, the viscosity of the normal breast epithelial cell was higher than that of cancer cells. For example, MCF-10A exhibited higher viscosity than MCF-7 cells. Moreover, MDA-MB-231 cells were less viscous than MCF-7, which indicates that when breast cancer cells are more invasive they are also less viscous [41].

Furthermore, the diameter of the fiber materials used for the preparation of the cell culture scaffold influenced the EMT phenomena. MDCK cells grew on PCL fiber scaffolds of 0.5 μm cells to form colonies with an epithelial phenotype. On the other hand, cells that grew on 5 μm scaffolds appeared more fibroblastic with a phenotype reminiscent of EMT [42]. It is known that mechanical, chemical, and geometrical cues collectively regulate the EMT process. A recent study showed that the photoactivatable gel substrate made of poly(acrylamide) (PAAm) hydrogel functionalized with poly-D-lysine (PDL) and photocleavable poly(ethylene glycol) (PEG) regulated EMT by altering the adsorption amount of ECM protein [43]. Another study showed that low energy protons (5 MeV) can enhance TGFβ1-induced EMT in non-transformed mink lung epithelial cells (Mv1Lu) and hTERT- immortalized human esophageal epithelial cells (EPC) [44]. On the other hand, it is unknown how material fluidity directs EMT. We cultured MCF-7 cells on crosslinked P(CL-*co*-DLLA) (solid-like) and non-crosslinked P(CL-*co*-DLLA) (liquid-like)100 and 500 substrates. Our findings showed that the cells which grew on non-crosslinked P(CL-*co*-DLLA) 500 substrates had an increased expression of E-cadherin and a decreased expression of vimentin. This suggests that the cells on non-crosslinked P(CL-*co*-DLLA) 500 substrates underwent the EMT phenomena. This could also be confirmed by the expression of E-cadherin and vimentin mRNA. In contrast, cells on non-crosslinked P(CL-*co*-DLLA) 100 substrates maintained their epithelial phenotype.

The principal signaling pathways which induce EMT are TGF-β, Wnt and Notch. Many EMT pathways control the transcription factors of the Snail family and the ZEB family [45]. TGF-β signaling plays a significant role in EMT induction both during normal development and in diseased states [46,47]. On the other hand, different TGF-β superfamily members are associated with driving EMT in murine mammary epithelial cells [48]. Increased matrix stiffness leads to contractile forces which enables the activation of TGF-β [49]. We also checked the expression of TGF-β mRNA. Our data showed that the epithelial cells transformed into mesenchymal cells which exhibited the upregulation of TGF-β mRNA, and was observed in both crosslinked P(CL-*co*-DLLA) and non-crosslinked P(CL-*co*-DLLA) 500 substrates. Although a better understanding of the molecular mechanism accounting for the activation and regulation of EMT is essential, the fluidity of the material is considered to be a crucial factor in determining cellular fate.

## 4. Materials and Methods

### 4.1. Preparation of Non-Crosslinked P (CL-co-DLLA) Substrates

The synthesis of four-branched copolymers of ε-caprolactone and D,L, lactide (DLLA) of 60:40 ratio of 100 or 500 unit length was performed as described in our earlier reports [21,22]. The molecular weights and structure of the polymers were estimated by GPC (JASCO International, Tokyo, Japan) and ^1^H NMR spectroscopy (JEOL, Tokyo, Japan), respectively. The viscoelastic properties of non-crosslinked P(CL-*co*-DLLA) substrates were tested using a rheometer (MCR 301, Anton Paar, Tokyo, Japan) and a viscoelastic spectrum which measured storage modulus (G’) and loss modulus (G’’).

Four-branched P(CL-*co*-DLLA) 100 or 500 units were dissolved in toluene at different concentration ranges from 1 *w/v%* to 100 *w/v%*. The solutions were then mixed with a small proportion of nile red (TCI America) and spin coated on 15 mm glass coverslip at 15,000 rpm for 60 sec. For cell culture, four-branched P(CL-*co*-DLLA) 100 or 500 units were dissolved in toluene at 80 *w/v%* and 50 *w/v%* respectively. The prepared P(CL-*co*-DLLA) substrates were then spin-coated on glass coverslips of 24-well inserts (15 mm) or 6-well inserts (35 mm) as described above and sterilized using a low-pressure hydrogen peroxide gas plasma system CH-160C (Toho Seisakusho, Tokyo, Japan).

### 4.2. Preparation of Crosslinked P(CL-co-DLLA) Substrates

The crosslinked P(CL-*co*-DLLA) substrates were prepared according to previous reports [21,22]. The mechanical property of the crosslinked P(CL-*co*-DLLA) substrates was characterized by a tensile test (EZ-S 500N; Shimadzu). The water contact angle was measured using a contact angle meter (VCA-Optima-Xe, AST, MA, USA). Scanning electron microscope (SEM) images of crosslinked and non-crosslinked P(CL-*co*-DLLA) substrates were examined with SU-8000 (Hitachi, Japan). For cell culture, the substrates were cut into the size of 24-well insets (15mm) or 6-well inserts (35mm).

### 4.3. Cell Culture

Human breast epithelial adenocarcinoma cells (MCF-7; ATCC) were cultured in minimum essential medium eagle’s (MEM) supplemented with 10% fetal bovine serum (FBS; ATCC), 0.1 mM non-essential amino acid (NEAA), 1 mM sodium pyruvate, and 1% antibiotic- antimycotic (anti-anti, Gibco). The cells were maintained in a humidified atmosphere of 5% CO_2_ at 37 °C. Before cell culture, the cover glass and P(CL-*co*-DLLA) (crosslinked and non-crosslinked) substrates were coated with 10 µg/mL of fibronectin (Sigma-Aldrich) for 1 h and excess fibronectin was washed by phosphate buffered saline (PBS; Gibco).

### 4.4. Immunostaining and Confocal Microscopy

The spin coated non-crosslinked P(CL-*co*-DLLA) 100 or 500 unit substrates of different concentrations (1 *w/v*% to 100 *w/v*%) were visualized under Leica SP5 confocal laser scanning microscope (Leica, Wetzlar, Germany) and the 3D images were scanned.

For immunostaining, MCF-7 cells were seeded at a density of 1 × 10^4^ cells/cm^2^ on a glass coverslip or P(CL-*co*-DLLA) (crosslinked and non-crosslinked) substrates and incubated at different time periods for up to three days. After the preferred incubation period, the cells were fixed by 4% paraformaldehyde (PFA; Wako, Japan) and blocked by 1% bovine serum albumin (BSA; Sigma-Aldrich) (in PBS) for 30 min. The cells were stained independently with monoclonal anti-vinculin (1:200; Sigma-Aldrich), anti-E-cadherin (1:100; glyceraldehyde-3-phosphate dehydrogenase glyceraldehyde-3-phosphate dehydrogenase BD biosciences) and anti-vimentin (1:100; Sigma-Aldrich) antibodies specific to vinculin, E-cadherin, and vimentin, respectively, followed by staining with Alexa Fluor^®^ 488 antimouse secondary antibody (1:100; Invitrogen). F-actin and nuclei were counterstained by tetramethylrhodamine B isothiocyanate-conjugated phalloidin (Sigma-Aldrich) and 4’,6-diamidino-2-phenylindole (DAPI; Sigma-Aldrich) respectively. The images were taken with a Leica SP5 confocal laser scanning microscope. The areas of the cell were measured by ImageJ (NIH, Maryland, USA).

### 4.5. Cell Proliferation and Adhesion Assay

MCF-7 cells were seeded on P(CL-*co*-DLLA) (crosslinked and non-crosslinked) substrates of 24- well plate inserts at a density of 1*10^4^ cells/cm^2^. The cells were incubated at different time periods from 1 to 5 days. A cell proliferation assay was performed using cell counting kit-8 (CCK-8; Dojindo, Japan) according to the manufacturer’s instructions. The cell adhesion assay was performed for 4 h and the relative cell adhesion was determined from the initial cell seeding density.

### 4.6. Quantitative Real-Time Polymerase Chain Reaction (qRT-PCR)

For the quantitative determination of mRNA expression, the qRT-PCR analysis was performed. 5 × 10^5^ cells/mL of MCF-7 cells were cultured on crosslinked and non-crosslinked P(CL-*co*-DLLA) substrates. The total cellular RNA was extracted using the RNeasy Kit (Qiagen) and extracted RNA was treated with DNaseI (Takara) to remove DNA contamination. Fifty ng of total RNA was reverse- transcribed into cDNA using the PrimeScript^TM^ 1^st^ strand cDNA synthesis kit (Takara) according to the manufacturer’s protocol. The qRT-PCR was performed using Light Cycler 1.5 System (Roche). The reaction mixture was composed of FastStart DNA Master^plus^ SYBER GREEN I, 20 µM of forward primer, 20 µM of reverse primer, and 1 µL of cDNA which made up a total volume of 15 μL which was treated with DEPC water. The thermocycling conditions were 95 °C for 10 s, followed by 45 cycles of 95 °C for 10 s, 60 °C for 7 s, and 70 °C for 10 s. The relative change in gene expression was quantified by threshold cycle (Ct), 2^−ΔΔCt^ method. Data were normalized with glyceraldehyde-3-phosphate dehydrogenase (GAPDH) as an endogenous control in the same reaction as the gene of interest. The primers used in this study are in Table 1.

### 4.7. Statistical Analysis

All results were represented as mean ± standard deviation (SD) obtained from three independent experiments. The degree of significance of each data was performed by Student’s t-test, where *p* < 0.05 is considered statistically significant when comparing non-crosslinked P(CL-*co*-DLLA) to crosslinked P(CL-*co*-DLLA) substrates.

## 5. Conclusions

EMT is a series of biochemical events in which polarized epithelial cells are transformed to a mesenchymal phenotype. In this study, we have prepared non-crosslinked (fluidic or viscous) material made of P(CL-*co*-DLLA) 100 and 500 units to understand the effect of materials fluidity on the EMT process. Initially, we checked the adhesion behavior and the expression of the focal adhesion protein of MCF-7 cells on these substrates. The adhesion of MCF-7 cells was enhanced by coating the surface of the substrates with fibronectin. The cells on non-crosslinked P(CL-*co*-DLLA) 500 substrates expressed a visible focal adhesion protein. However, cells on non-crosslinked P(CL-*co*-DLLA) 100 substrates did not show a visible focal adhesion. This could correlate with the proliferation behavior and EMT phenomenon of the cells on non-crosslinked P(CL-*co*-DLLA) 100 and 500 substrates. Cell proliferation increased with an increasing culture period on non-crosslinked P(CL-*co*-DLLA) 500 substrates. On the other hand, the proliferation decreased by increasing the culture period on non-crosslinked P(CL-*co*-DLLA) 100 substrates. EMT was observed with a decrease in the epithelial marker and an increase in the mesenchymal marker. We found the cells underwent EMT in non-crosslinked P(CL-*co*-DLLA) 500 substrates by a gradual increase in the expression of vimentin. At 72 h, about 100% *o*f the cells underwent EMT. The principal signaling pathway which induced EMT was TGF-β. The expression of TGF-β also increased with increased time on non-crosslinked P(CL-*co*-DLLA) 500 substrates. This also suggests that the induction EMT process is associated with TGF-β. In contrast, the cells maintained their epithelial phenotype on non-crosslinked P(CL-*co*-DLLA) 100 substrates by maintaining the expression of E-cadherin. Our findings show that material fluidity is a crucial factor to determine the EMT process. Further, it is possible to regulate the degree of EMT by altering the fluidity of the material.

## Figures and Tables

**Figure 1 ijms-21-01757-f001:**
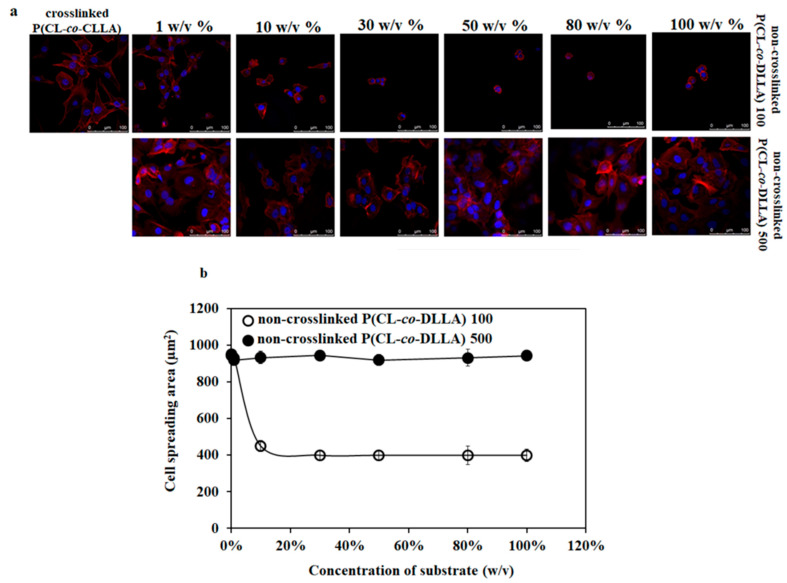
(**a**) Confocal microscopic images of cell spreading behavior of MCF-7 cells on crosslinked and non-crosslinked P(CL-*co*-DLLA) 100 or 500 substrates of different concentrations (1–100 *w/v%*). The spreading behavior of MCF-7 cells decreases with an increased concentration of non-crosslinked P(CL-*co*-DLLA) 100 substrates while the morphology of MCF-7 cells remains unchanged on any concentration of non-crosslinked P(CL-*co*-DLLA) 500 substrates. (**b**) Graphical representation of the area of cells on crosslinked and non-crosslinked P(CL-*co*-DLLA) substrates. The cell spreading area decreases when the concentration of P(CL-*co*-DLLA) 100 substrates (white circles) increases and the cells remain in the elongated morphology in any concentration of P(CL-*co*-DLLA) 500 substrates (black circles). Fifty cells were counted for each sample and are represented with ± SD.

**Figure 2 ijms-21-01757-f002:**
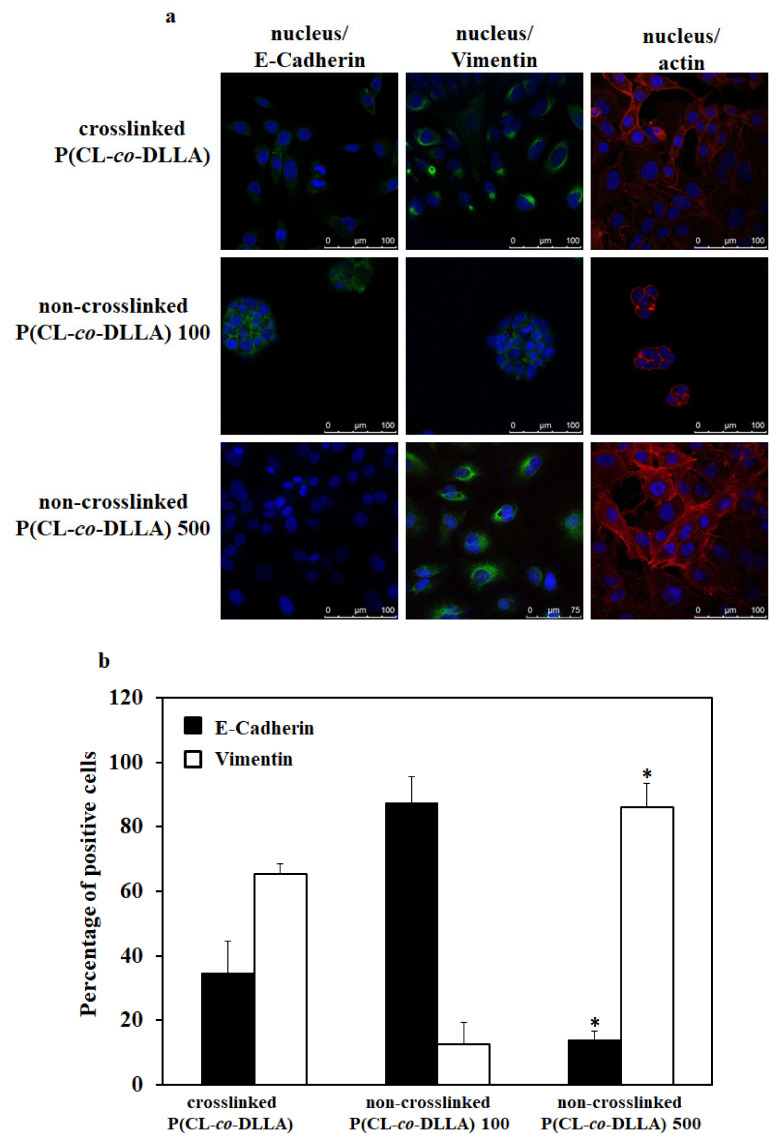
Biochemical evaluation of epithelial (E-cadherin) and mesenchymal (vimentin) markers in MCF-7 cells cultured on P(CL-*co*-DLLA) (crosslinked and non-crosslinked) substrates. The images represent the expression of E-cadherin and vimentin in MCF-7 cells at (**a**) 48 h and (**c**) 72 h. MCF-7 cells on non-crosslinked P(CL-*co*-DLLA) 100 substrates maintained their epithelial phenotype throughout the culture period whereas MCF-7 cells on non-crosslinked P(CL-*co*-DLLA) 500 substrates transformed to a mesenchymal phenotype. Quantitative determination of the percentage of E-cadherin (black bars) and vimentin (white bars) positive cells (**b**) 48h and (**d**) 72 h. In the crosslinked P(CL-*co*-DLLA) substrate, about 65% of cells were transformed to mesenchymal cells at 48 h and about 75% of epithelial cells were transformed to mesenchymal cells at 72 h. In the non-crosslinked P(CL-*co*-DLLA) 500 substrate, at 48 h, 85% of MCF-7 cells and at 72 h around 96% of the cells were transformed to mesenchymal phenotype. MCF-7 cells on non-crosslinked P(CL-*co*-DLLA) 100 substrates maintained their epithelial phenotype throughout the culture period. (**e**) Time-dependent quantitative representation of E-cadherin and (**f**) vimentin mRNA expression by quantitative real-time PCR (qRT-PCR) where glyceraldehyde-3-phosphate dehydrogenase (GAPDH) is used as an endogenous control of the same sample of each experiment. *n* = 3 ± SD, * *p* = 0.05 and ** *p* = 0.01 for non-crosslinked P(CL-*co*-DLLA) substrate corresponding to crosslinked P(CL-*co*-DLLA) substrate of the same experiment.

**Figure 3 ijms-21-01757-f003:**
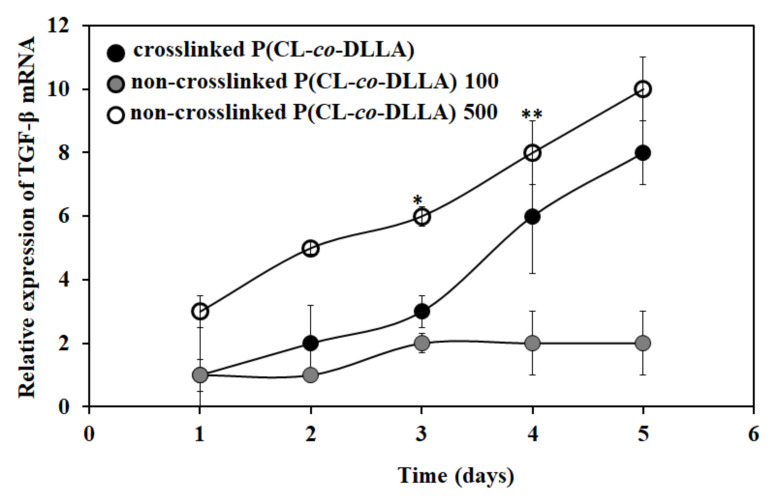
Relative expression of TGF-β mRNA in MCF-7 cells on P(CL-*co*-DLLA) (crosslinked and non-crosslinked) substrates at different time periods. The expression of TGF-β mRNA is increased in crosslinked P(CL-*co*-DLLA) and non-crosslinked P(CL-*co*-DLLA) 500 substrates. The data were normalized to GAPDH as the exogenous control. Each data represents *n* = 3 ± SD, where * *p* = 0.05 and ** *p* = 0.01 for non-crosslinked P(CL-*co*-DLLA) substrate regards to crosslinked P(CL-*co*-DLLA) substrate.

**Figure 4 ijms-21-01757-f004:**
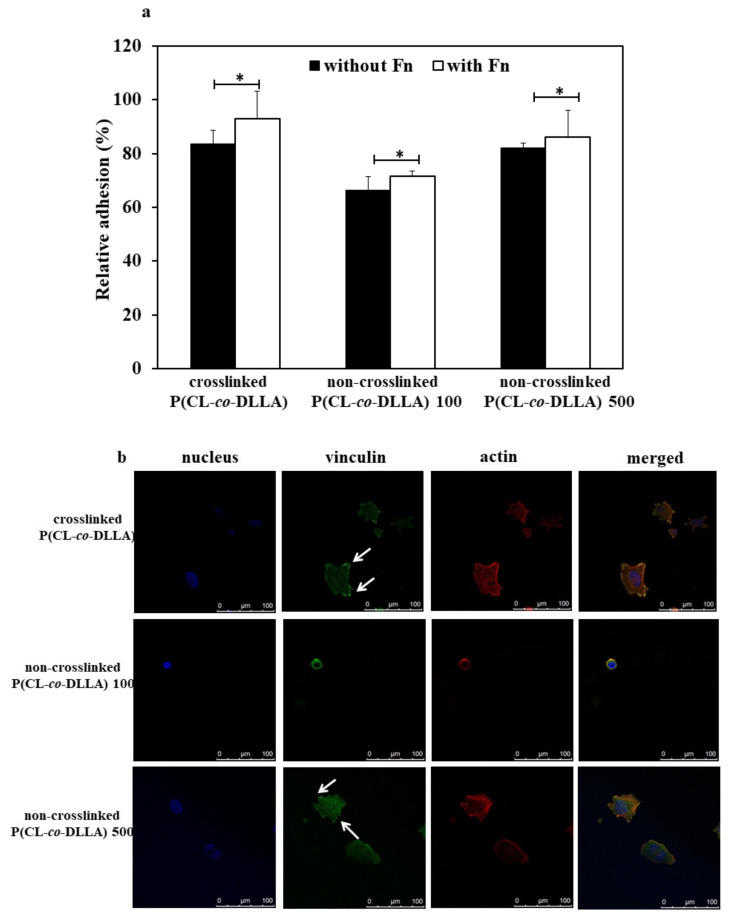
Adhesion behavior of MCF-7 cells on crosslinked and non-crosslinked P(CL-*co*-DLLA) substrates. (**a**) The substrates were coated with or without fibronectin (Fn) and the percentage of attached cells were calculated from the initial cell seeding density after 4 h. Each data represent *n* = 3 ± SD, where * *p* = 0.05 and ** *p* = 0.01; (**b**) Focal adhesion of MCF-7 cells on P(CL-*co*-DLLA) crosslinked (top images) and P(CL-*co*-DLLA) non-crosslinked 100 (middle images) or 500 (bottom images) substrates at 90 min. The white arrows indicated the focal adhesion protein vinculin.

**Figure 5 ijms-21-01757-f005:**
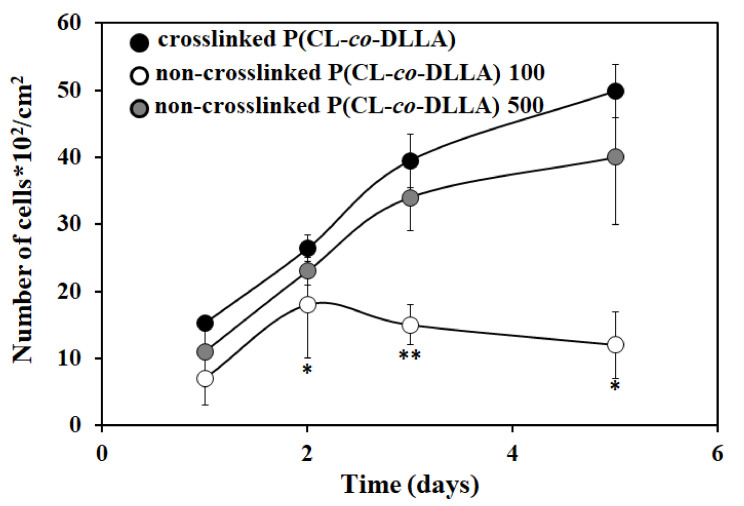
Proliferation behavior of MCF-7 cells on crosslinked P(CL-*co*-DLLA) non-crosslinked P(CL-*co*-DLLA) substrates at different time periods (1 day to 5 days). The cell proliferation increased by increasing the culture period on non-crosslinked P(CL-*co*-DLLA) 500 substrate. On non-crosslinked P(CL-*co*-DLLA) 100 substrates, initially the proliferation increased and later on the proliferation decreased. Each data represent *n* = 3 ± SD, where * *p* = 0.05 and ** *p* = 0.01 compared to crosslinked P(CL-*co*-DLLA) substrates.

**Table 1 ijms-21-01757-t001:** List of oligonucleotides used for quantitative analysis of gene expression by quantitative real-time PCR (qRT-PCR).

Genes	Forward Primer (5′-3′)	Reverse Primer (5′-3′)
GAPDH	CCCCCACCACACTGAATCTC	GCCCCTCCCCTCTTCAAG
E-cadherin	AACGCATTGCCACATACACTC	GACCTCCATCACAGAGGTTCC
vimentin	TCAATGTTAAGATGGCCCTTG	TGAGTGGGTATCAACCAGAGG
TGF-β	CCCAGCATCTGCAAAGCTC	GTCAATGTACAGCTGCCGCA

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
