# Peer review of "Fluidity of Poly (ε-Caprolactone)-Based Material Induces Epithelial-to-Mesenchymal Transition"

_ijms, 2020, doi:10.3390/ijms21051757_

Round 1

Reviewer 1 Report

Dr. Mano et al. addressed a report with the aim of studying the effect of material fluidity (using in particular poly (ε-caprolactone-co-D, L-lactide) with tunable elasticity and fluidity) on EMT. As correctly mentioned be the authors, EMT is a complex process that occurs through many events such as transcription factors activation, specific cell surface proteins expression, reorganization and expression of cytoskeletal proteins, regulation of ECM proteins, degradation of cell to cell adhesion, regulation in the expression of specific miRNAs, and an increased ability to migrate and invade. However, few mechanisms have been considered in this work (E-cadherin, vimentin and TGF-beta expression) without studying at least some of key EMT-transcription factors (i.e. SNAIL, ZEB1, JUNB, TWIST, JAK/STAT3), other EMT-markers (i.e. N-cadherin, alpha-SMA) and the increased ability of epithelial cells to migrate and invade during EMT. Althoug the authors gave promising data showing that their system could be used to regulate the degree of EMT by turning the fluidity of the material, there are some issues need to be addressed:

The title should be changed to emphasize a POTENTIAL application of material fluidity to induce EMT; In the abstract, the first sentence (line 17-18) should be changed emphasizing the POTENTIAL application of material fluidity to induce EMT; I suggest that the authors modify the introduction: the introduction should better delve into the advantages that lead to the use of (P(CL-co-DLLA) as experimental material for studying in-vitro the EMT process reducing the discussion on the results (page 2, line 60-75);   An migration/invasion analysis on MCF7 should be done to confirm EMT phenotype; On page 2, line 58, please delete “that”; On page 2, line 62, please delete “were”; On page 2, line 66, please delete “that”; The figures are not well presented including label and quality; Please add figure legend in figure 1b, 2b, 2d; On page 4, line 125, the title “Effect of Non-crosslinked P(CL-co-DLLA) Substrates on Epithelial-to- Mesenchymal Transition (EMT)” should be changed with “Effect of Non-crosslinked P(CL-co-DLLA) Substrates on E-cadherin and vimentin expression”; On page 7, line 174, the title “Effect of Substrate Fluidity on Epithelial-to-Mesenchymal Transition (EMT)” should be changed with “Effect of Substrate Fluidity on TGF-β expression”; On page 8, line 197, please add “.” After “(Figure 4a, white bars)”; On page 9, line 218-223, the description does not correspond with the figure. Please clarify; On page 10, line 256, please change “(EndMT)” with “(EMT)”; On page 13, line 381, pleas delete “that”; Figs s4 and 2 should be unified; Please, in fig s3 add scale bar in all images.

Author Response

Reviewer #1

Q1. The title should be changed to emphasize a POTENTIAL application of material fluidity to induce EMT

A1. We would like to thank you for your precious time. As the reviewer suggested, we conducted preliminary studies on material fluidity to induce EMT process. However, this study demonstrates the first studies on material fluidity to induce EMT process. So, if possible, we would like to keep our title “Fluidity of poly (ε-caprolactone)-based material induces epithelial-to-mesenchymal transition”. Instead, we modified the abstract and introduction in order to emphasize a potential application of material fluidity to induce EMT.

Q2. In the abstract, the first sentence (line 17-18) should be changed emphasizing the POTENTIAL application of material fluidity to induce EMT.

A2. Thank you very much for your kind suggestion. According to the reviewer’s suggestion, we mentioned in the abstract “potential studies on material fluidity to induced epithelial to mesenchymal transition (EMT) in MCF-7 cells” (Page 1, lines 17-18).

Q3. An migration/invasion analysis on MCF7 should be done to confirm EMT phenotype;

A3. Thank you very much for your kind suggestion. As the reviewer suggested cell migration analysis is one of the important assay to consider for EMT process. However, we had conducted preliminary experiments on materials fluidity to induce EMT in this manuscript, we are focusing on cell migrations in fluidic materials and the effects of materials fluidity in EMT further.

Q4. I suggest that the authors modify the introduction: the introduction should better delve into the advantages that lead to the use of (P(CL-co-DLLA) as experimental material for studying in-vitro the EMT process reducing the discussion on the results (page 2, line 60-75).

A4. Thank you for your suggestion. As reviewer’s suggestion we carefully checked the introduction section and the introduction part has been modified by reducing the discussion part and emphasizing the advantages of fluidic P(CL-co-DLLA) material in the introduction. (Page 2, lines 62-75 and page 3, lines 77-79).

Q5. On page 2, line 58, please delete “that”

A5. Thank you very much for your comment. As reviewer suggested we had deleted “that” from Page 2, line 58.

Q6. On page 2, line 62, please delete “were”

A6. Thank you very much for your comment. As reviewer suggested we had deleted “were” - Page 2, line 70

Q7. On page 2, line 66, please delete “that”

A7. Thank you very much for your comment. As reviewer suggested we had deleted “that” - Page 2, line 71.

Q8. The figures are not well presented including label and quality

A8. Thank you for your comments regarding the figure. As reviewer suggested, we carefully checked each figure and the quality of each figure has been improved along with the markings of the figure.

Q9. Please add figure legend in figure 1b, 2b, 2d.

A9. Thank you for your kind suggestion regarding figure legends. As reviewer’s comments we had carefully checked each figure caption and included the explanation for each figures such as figures 1b- Page4, lines 123-126, 2b and 2d- page 8, lines 172-177.

Q10. On page 4, line 125, the title “Effect of Non-crosslinked P(CL-co-DLLA) Substrates on Epithelial-to- Mesenchymal Transition (EMT)” should be changed with “Effect of Non-crosslinked P(CL-co-DLLA) Substrates on E-cadherin and vimentin expression”

A10. Thank you very much for your suggestion. We had changed Effect of Non-crosslinked P(CL-co-DLLA) Substrates on Epithelial-to- Mesenchymal Transition (EMT)” to “Effect of Non-crosslinked P(CL-co-DLLA) Substrates on E-Cadherin and Vimentin expression” (Page 4, line 128) as reviewer’s comment.

Q11. On page 7, line 174, the title “Effect of Substrate Fluidity on Epithelial-to-Mesenchymal Transition (EMT)” should be changed with “Effect of Substrate Fluidity on TGF-β expression”

A11. Thank you very much for your suggestion. We had changed Effect of Substrate Fluidity on Epithelial-to-Mesenchymal Transition (EMT) to “Effect of Substrate Fluidity on TGF-β expression” (Page 9, line 182) as reviewer’s comment.

Q12. On page 8, line 197, please add “.” After “(Figure 4a, white bars)”.

A12. Thank you very much for your suggestion. We had added Full stop “.” in Page 10, line 208 after Figure 4a, white bars as reviewer’s comment.

Q13. On page 9, line 218-223, the description does not correspond with the figure. Please clarify.

A13. Thank you very much for your comment. We are pleased to inform you that we had mistakenly input the figure marking in Figure 5. So we had changed the figure with appropriate marking such as white circle as non-crosslinked P(CL-co-DLLA) 100 substrate and grey circle as P(CL-co-DLLA) 500 substrates. Page 12, Figure 5.

Q14. On page 10, line 256, please change “(EndMT)” with “(EMT)”

A14. Thank you very much for your comment. We mentioned the reference 39 (Zhang, H.; Chang, H.; Wang, L.; Ren, K.; Martins, M.C.; Barbosa, M.A.; et al. Effect of polyelectrolyte film stiffness on endothelial cells during endothelial-to-mesenchymal transition. Biomacromolecules. 2015, 16, 3584-3593) in discussion section described about the effect of material stiffness on endothelial to mesenchymal transition. Thus in page 13, line 267 it is mentioned as EndMT instead of EMT.

Q15. On page 13, line 381, pleas delete “that”.

A15. Thank you very much for your comment. We had removed “that” from Page 16, lines- 39.

Q16. Figs s4 and 2 should be unified

A16. Thank you very much for your comment. As reviewr’s commet to show the uniformity in the figure, Figure S4 (Supplementary- Pages 4-5) and Figure 2 (Pages 6-8) were unified.

Q17. Please, in fig s3 add scale bar in all images.

A17. Thank you very much for your comment. We had carefully checked each image in Figure S3 and scale bar is included for all the images in Figure S3. (Supplementary- Pages 4).

Reviewer 2 Report

The authors state that epithelial to mesenchymal transition (EMT) as a major biological process occurs during embryogenesis, wound healing, fibrosis and cancer. As they mentioned, the process occurs through a series of events. The expression of Vimentin, E-Cadherin and Vinculin is essential but not the singular determining factor to declare the onset of offset of EMT. Please consider rewriting your general statement. 

Description of Fig.2 b+d should be improved. Please declare Vimentin and E-Cadherin in the Figure legends. Exact p=values are preferred in all figures.

The influence of TGF beta should be discussed. Is the EMT-like shift a result of the fluidity or of an increased TGF beta on the scaffolds. This may present a limitation of the study and should be addressed.

Please revise part 2.5. Effect of Non-crosslinked P(CL-co-DLLA) Substrate on Cell Proliferation. ...The cell proliferation is increased by increasing the culture period on non-crosslinked P(CL-co-DLLA) 500 substrates. On non-crosslinked P(CL-co-DLLA) 100 substrate in contrast, initially the proliferation is increased until 2 days. Later on from day 3, the proliferation is decreased.... It appears that the corresponding figure indicated a cellular derease in P(CL-co-DLLA) 500.

  On the other hand, it is unknown that how the material fluidity directly involved in the 58 regulation of EMT.

Author Response

Reviewer #2

Q1. The authors state that epithelial to mesenchymal transition (EMT) as a major biological process occurs during embryogenesis, wound healing, fibrosis and cancer. As they mentioned, the process occurs through a series of events. The expression of Vimentin, E-Cadherin and Vinculin is essential but not the singular determining factor to declare the onset of offset of EMT. Please consider rewriting your general statement.

A1. We would like to thank you for your precious time. As the reviewer suggested, we only show preliminary data in this manuscript. We carefully checked the introduction section and the introduction part have been modified. We had included more references in this sections (ref: 17-20). Page.2, lines 62-69

Q2. Description of Fig.2 b+d should be improved.

A2. Thank you for your kind suggestion regarding figure legends. As reviewer’s comments we had carefully checked each figure caption and included the explanation for each figures such as figures 2b and 2d- page 8, lines 172-177.

Q3. Please declare Vimentin and E-Cadherin in the Figure legends.

A3. Thank you very much for your kind suggestion. As reviewer’s suggestion, we carefully checked our manuscript and “vimentin” and “E-cadherin” were declared as “Vimentin” and “ E-Cadherin” in Fig. 2 (Pages 6-7, Page 8, lines-166,168,178), Supplementary fig S4 (Page 3) and throughout the manuscript (Page 1-lines 25-26, Page 2- lines 72-73, Page 4- line 133, Page 5- line-139, 150-154, Page 13- line 269-270, 289,291, Page 14-lines 346,347, Table1-Page 15, Page 16-lines 395,400)

Q4. Exact p=values are preferred in all figures.

A4. Thank you very much for your comment. We carefully check the manuscript as per reviewer’s comments, specific p- value are mentioned in all the figures and supplementary figures, Fig 2 (Page 8, line- 179, Page 9- 180), Fig 3 (Page-9, line- 195, 196), Fig 4 (Page 11, line-224), Fig 5 (Page 12, line- 240) Supplementary fig S4 (Page 5).

Q5. The influence of TGF beta should be discussed. Is the EMT-like shift a result of the fluidity or of an increased TGF beta on the scaffolds. This may present a limitation of the study and should be addressed.

A5. Thank you for your kind suggestion. In our experiments, we didn’t treat our materials with TGF-β. We only analyzed the expression of TGF- β mRNA in cells cultured in different P(CL-co-DLLA) substrates. The cells underwent EMT showed increased expression of TGF-β. We added a reference (Ref: 26) mentioned the important of TGF-β in EMT process (Page 9, lines 183-185).

Q6. Please revise part 2.5. Effect of Non-crosslinked P(CL-co-DLLA) Substrate on Cell Proliferation. ...The cell proliferation is increased by increasing the culture period on non-crosslinked P(CL-co-DLLA) 500 substrates. On non-crosslinked P(CL-co-DLLA) 100 substrate in contrast, initially the proliferation is increased until 2 days. Later on from day 3, the proliferation is decreased.... It appears that the corresponding figure indicated a cellular decrease in P(CL-co-DLLA) 500. There was a mistake in the figure.

A6. Thank you very much for your comment. We are pleased to inform you that we had mistakenly input the figure marking in Figure 5. So we had changed the figure with appropriate marking such as white circle as non-crosslinked P(CL-co-DLLA) 100 substrate and grey circle as P(CL-co-DLLA) 500 substrates. Page 12, Figure 5.

Round 2

Reviewer 1 Report

  • On page 2 line 62-67, the sentence “EMT is a crucial process in numerous physiological processes such as early embryonic development, mammary gland development, myofibroblast formation during fibrosis as well as many stages of tumor progression [17]. Several transcription factors such as ZEB, TWIST, SNAIL, members of FOX family and GATA regulate [18] EMT processes by promoting the loss of cell-to-cell adhesion, variation in cytoskeletal arrangement and transition from epithelial phenotype to mesenchymal phenotype [19]” is redundant with a previous sentence “EMT is a major physiological process that occurs during embryogenesis, wound healing, fibrosis as well as in cancer metastasis [7]. EMT process occurs through a series of events such as transcription factors activation [8], specific cell surface proteins expression [9], reorganization and expression of cytoskeletal proteins [10], regulation of ECM proteins [11], degradation of cell to cell adhesion [12], regulation in the expression of specific miRNAs [13], and an increased ability to migrate and invade [6]”. In the page 2 line 62-67, the authors should detailed the role of E-cadherin and vimentin in the EMT process.

  • Please add figure legends in figure 1b [What do the white and black circles represent? The author should specify in the picture what the white or black circle represents: P(CL-co-DLLA) 500 or P(CL-co-DLLA) 100 substrates]. 2b and 2d [the same description should be made for white bar (vimentin or cadherin) and black bar (vimentin or cadherin)].

Author Response

Q1. On page 2 line 62-67, the sentence “EMT is a crucial process in numerous physiological processes such as early embryonic development, mammary gland development, myofibroblast formation during fibrosis as well as many stages of tumor progression [17]. Several transcription factors such as ZEB, TWIST, SNAIL, members of FOX family and GATA regulate [18] EMT processes by promoting the loss of cell-to-cell adhesion, variation in cytoskeletal arrangement and transition from epithelial phenotype to mesenchymal phenotype [19]” is redundant with a previous sentence “EMT is a major physiological process that occurs during embryogenesis, wound healing, fibrosis as well as in cancer metastasis [7]. EMT process occurs through a series of events such as transcription factors activation [8], specific cell surface proteins expression [9], reorganization and expression of cytoskeletal proteins [10], regulation of ECM proteins [11], degradation of cell to cell adhesion [12], regulation in the expression of specific miRNAs [13], and an increased ability to migrate and invade [6]”. In the page 2 line 62-67, the authors should detailed the role of E-cadherin and vimentin in the EMT process.

A1. We would like to thank you for your precious time. As the reviewer suggested, we carefully checked the introduction section of the manuscript and deleted the repeated sections. The role of E-cadherin and Vimentin in EMT has been emphasized in Page-2, lines- 63-68. New references (ref: 17-19) were cited which described the role of E-cadherin and Vimentin in Page 17, ref 17-19.

Q2. Please add figure legends in figure 1b [What do the white and black circles represent? The author should specify in the picture what the white or black circle represents: P(CL-co-DLLA) 500 or P(CL-co-DLLA) 100 substrates]. 2b and 2d [the same description should be made for white bar (vimentin or cadherin) and black bar (vimentin or cadherin)].

A2. Thank you for your kind comment regarding figure legends. As reviewer’s comments we had included the expansion of  white circles and black circles for Figure 1b in Page 4 and in the figure legends such as  white circles which represent P(CL-co-DLLA) 100 substrates and black circles which represent P(CL-co-DLLA) 500 substrates in Page 4, lines 126 and 128. In Figure 2b and 2d the description for black bars and white bars added in Figure 2b, page 6 and Figure 2d, page7. The expansion of white bars and black bars were also included in figure legends of Figure 2b and 2d in Page8, line 173.